# Challenges and Strategies to Optimising the Quality of Small Bowel Magnetic Resonance Imaging in Crohn’s Disease

**DOI:** 10.3390/diagnostics12102533

**Published:** 2022-10-19

**Authors:** Anuj Bohra, Abhinav Vasudevan, Numan Kutaiba, Daniel R. Van Langenberg

**Affiliations:** 1Department of Gastroenterology, Eastern Health, Box Hill 3128, Australia; 2Department of Radiology, Eastern Health, Box Hill 3128, Australia

**Keywords:** magnetic resonance enterography, Crohn’s disease, diagnostics, bowel distension, motion artifact

## Abstract

Magnetic resonance enterography (MRE) is one of the most highly utilised tools in the assessment of patients with small bowel Crohn’s disease (CD). As a non-invasive modality, it has both patient and procedure-related advantages over ileocolonoscopy which is the current gold standard for Crohn’s disease activity assessment. MRE relies upon high-quality images to ensure accurate disease activity assessment; however, few studies have explored the impact of image quality on the accuracy of small bowel CD activity assessment. Bowel distension and motion artifacts are two key imaging parameters that impact the quality of images obtained through MRE. Multiple strategies have been employed to both minimise the effects of motion artifacts and improve bowel distension. This review discusses the definitions of bowel distension and motion artifacts within the literature with a particular focus on current strategies to improve bowel distension and limit motion artifacts in MRE.

## 1. Introduction

Crohn’s disease (CD) is a chronic immune-inflammatory condition that can affect the entire gastrointestinal tract (GIT), with over 40% of patients exhibiting ileal involvement [1]. Small bowel CD can be assessed via multiple modalities, such as ileocolonoscopy, diagnostic imaging with magnetic resonance enterography (MRE) and intestinal ultrasound (IUS) and/or serum and faecal biomarkers [2,3]. Whilst ileocolonoscopy remains the gold standard of inflammatory bowel disease (IBD) assessment, multiple factors limit its widespread and frequent utility. These include capacity constraints, procedural risks, onerous preparation requirements, and the inability to evaluate the proximal small bowel. With increasingly stringent treatment targets and the need for regular objective evaluations [4], there is a growing reliance on non-invasive modalities of disease activity assessment such as MRE and IUS.

The use of MRE for assessment of small bowel CD has been recommended by international CD guidelines, with normalisation of small bowel findings, including the achievement of transmural healing, seen as a potential treatment target [2,3,4]. This is based on the background of multiple sentinel studies showing high correlation of endoscopic CD disease activity compared with MRE [5,6] and multiple MRE indices of activity having been developed and validated in CD [7,8,9,10]. For instance, the simplified magnetic resonance index of activity (sMaRIA) score was recently developed from the original MaRIA score and is a relatively easy and standardised MRE reporting score in CD. This score and similar indices are a step forward in improving quality in MRE [8]. However, although smaller external validation studies have assessed MRE indices of CD activity, there remains uncertainty about their external validity due to variations in MRE protocols worldwide [11].

Typically, MRE protocols and image sequence acquisition are shaped by local expertise and manufacturer recommendations. It remains unclear whether subtle alterations in protocol produce significant variations in image quality. For example, in the original MaRIA and sMaRIA protocols, retrograde per-rectal saline instillation of up to 2000 mL via rectal catheter and standard colonoscopy bowel preparation regimens were utilised. These have been shown to improve bowel distention and/or reduce motion artifacts in MRE, but these are not a universal part of MRE protocols due to patient tolerability [7,8]. The Clermont score, which incorporates diffusion-weighted imaging (DWI), was developed from the original MaRIA [10]. Within the original study, the MRE protocol used to derive the Clermont score did not include retrograde saline instillation or bowel preparation and substituted intravenous (IV) hyoscine butylbromide for glucagon [10]. The absence of these factors when deriving the score was raised as a potential cause of underreporting of MRE findings [10]. To date, no MRE index of CD activity has sought to assess the impact of quality metrics on the accuracy of disease activity assessment.

The measurement and integration of quality metrics in healthcare has been shown to improve disease-related outcomes in multiple disease states, including CD [12,13]. Furthermore, the incorporation of quality metrics in other gastrointestinal investigations integral to IBD care, such as the Boston bowel preparation score for colonoscopy, has been shown to improve diagnostic accuracy [14,15]. Yet a quality metric for MRE, a vital investigation in the diagnosis and management of CD upon which life-changing clinical decisions are frequently made, does not exist [14,15]. Hence, this review article explores factors affecting the quality of MRE in CD with a focus on motion artifacts and bowel distension, the two most frequently encountered radiological artifacts in bowel imaging [16]. We also examine the literature with regards to defining the adequacy of bowel distention and motion artifacts and explore potential strategies to optimise these metrics in MRE for the assessment of CD.

## 2. Components of Magnetic Resonance Enterography/Enteroclysis

MRE/enteroclysis protocols have rapidly evolved since inception as a validated tool for the assessment of small bowel CD [17]. Worldwide, multiple MRE protocols exist, but a broad discussion of various image acquisition sequences is beyond the scope of this review and is presented elsewhere [18]. This article discusses the general aspects of typical protocols as outlined below.

### 2.1. Choice of MRI Machine/Software/Imaging Protocol

MRE studies are usually performed on both 1.5 or 3 Tesla (T) field strengths depending on local availability. There is no consensus on which field strength is optimal for MRE [19]. A study of 26 patients who underwent both 1.5 and 3 T MREs at the same time using ileocolonoscopy as the reference standard showed similar accuracy in evaluating ileo-colonic CD but superiority of 3 T for detection of mucosal ulcers [20]. Another study of 88 patients found no significant differences in accuracy between 1.5 and 3 T MREs [21]. Higher field strengths such as 7 T remain experimental in the setting of MRE and beyond current practice [22].

Sequences acquired for MRE routinely include T1- and T2-weighted imaging in axial and coronal planes. T2-weighted imaging can be acquired as balanced steady-state free precession or single-shot fast spin echo sequences. Both types of sequences are rapidly acquired and depict luminal fluid well. At least one of these sequences should be acquired with fat suppression to allow better depiction of mural edema when compared to adjacent fat (suppressed) [18,23]. Post-contrast T1-weighted imaging is routinely acquired unless there is a contraindication to gadolinium-based contrast agents. This sequence is usually acquired with fat suppression in coronal and axial planes and allows for assessment of wall enhancement as well as penetrating disease and fluid collections. While this sequence is susceptible to motion artifacts, it is usually acquired after an antispasmodic has been administered to minimize bowel motion artifacts [18]. Additional sequences such as DWI (particularly useful when no contrast imaging is available) and dynamic cine sequences (for peristalsis) have been suggested and are usually included in scanning protocols [19,23]. Several studies on DWI in MRE have shown heterogenous results [24]. Furthermore, apparent diffusion coefficient (ADC) measurements, which rely on multiple DWI b-values, are challenging in daily practice due to lack standardization of imaging acquisition, difficulties in reproducibility of ADC values and susceptibility to various artifacts [24,25,26]. Optimisation of MRI sequences depends on manufacturer software packages and recommendations as well as local expertise. For example, the choice of performing DWI with free-breathing versus breath holding versus navigator-triggered relies on the balance of imaging acquisition time and quality of obtained images [25]. Like MRI of other body parts, sequences acquired on machines from disparate manufacturers will differ slightly. However, multicentre studies demonstrating variability of performance of sequences from different manufacturers in CD disease activity assessment are lacking.

### 2.2. Fasting

Fasting for 4–6 h prior to the procedure is generally recommended [19]. It is postulated that an empty, non-distended stomach results in greater tolerance of high-volume oral contrast ingestion as is required pre-procedure. Moreover, fasting and in particular avoidance of foods that cause bloating prior to MRE is thought to reduce gas within the bowel which can be a source of artifacts particularly on DWI sequences [27]. A third important reason for fasting is to reduce food material in the small and large bowel which can appear hyperintense on T1-weighted images [27]. The presence of an increased T1 weighted signal can affect post-contrast bowel wall enhancement and subsequent CD activity detection [27]. In addition, fasting prevents bladder overfilling during the image acquisition period. A full bladder during imaging acquisition may result in a patient feeling the urge to void and result in motion artefact. In addition, a distended bladder reduces intra-abdominal volume occupied by bowel segments in the pelvis which may result in crowding of bowel loops and poor separation of bowel walls. Despite these theoretical considerations, to our knowledge, the advantages of fasting and the duration of fasting have not been directly studied in MRE.

### 2.3. Oral Contrast

Oral contrast ingestion is used to distend normally collapsed loops of small bowel whilst replacing luminal bowel content with a uniform intraluminal material. Traditionally, 1000–1500 mL is ingested over a period of 45 to 60 min prior to the scan [19]. This timeframe and volume of contrast vary slightly at different centres and may be reduced in patients with an ileostomy and/or shortened gut length from resections [19]. In MR enteroclysis, the oral contrast is delivered immediately prior to image acquisition via a nasoenteral tube.

Enteric contrast agents are traditionally biphasic, appearing dark on T1-weighted and bright on T2-weighted images [17]. This allows for appreciable mural contrast enhancement on T1 contrast enhanced sequences [28]. Whilst water can provide adequate bowel distention, it is rapidly absorbed in the jejunum, rendering it unsuitable for ileal distention. Thus, ideal contrast agents contain non-absorptive additives that retain water in the intraluminal space. Examples of additives include polyethylene glycol, mannitol, sorbitol, and low-density barium. However, a clear consensus on the ideal volume, timing, and type of oral contrast to achieve optimal bowel distention has not been established but is instead often dependent on local availability, experience, and patient tolerance. This is discussed further in later sections of this review.

### 2.4. Intravenous Contrast

MRE protocols usually include T1 sequences prior to and after intravenous (IV) gadolinium-based contrast to improve CD activity detection. Key findings in CD enhanced by IV contrast use include bowel wall enhancement, improved vasculature assessment, and the presence of lymph nodes, fistulas, and/or abscesses [17,29]. IV gadolinium contrast dosing is calculated based upon the patient’s weight and estimated glomerular filtration rate (eGFR). Contraindications include previous contrast allergy, pregnancy, renal impairment (eGFR ≤ 30), and/or peritoneal dialysis [17]. Due to the lack of an IV contrast requirement, DWI sequences are of particular importance in those with known IV-contrast-derived contradictions. A number of studies have evaluated the performance of MRE in IBD assessment without IV contrast using non-contrast TI, T2, and DWI sequences with similar accuracy seen comparative to post-contrast sequences [30,31,32]. Thus, an employable strategy may be to restrict IV contrast use in patients with known CD with suspected penetrating complications with screening MRE performed without IV contrast and emphasis placed on T2 and DWI sequences for disease activity assessment.

### 2.5. Antiperistalsis Agents

Normal small bowel peristalsis during MRE image acquisition results in motion artifacts with subsequent degradation in image quality [27]. Thus, antiperistalsis agents are routinely administered through various stages of MRE image acquisition, in particular those sequences with a longer image acquisition time [27]. Both hyoscine butylbromide and glucagon are widely used in this setting; however; glucagon is the only available agent in the USA, with both agents available in most other regions.

Hyoscine is administered intravenously (IV) in single or split dosing with 2–3 injections. Doses ranging between 10–40 mg have been reported in various international protocols without a clear improvement in motion artifacts seen with higher doses [11,19]. Hyoscine is generally avoided in patients with known glaucoma, myasthenia gravis, and tachyarrhythmias; thus, in these scenarios’ glucagon is preferred [27]. Glucagon is administered intravenously in single or split doses of between 0.5–1 mg based upon body weight [27]. Glucagon’s inherent properties have demonstrated a capacity to improve visualisation of the bowel wall within the terminal ileum through reduction in motion-related artifacts [33].

Studies have demonstrated potential benefits with intramuscular (IM) administration of both hyoscine and glucagon [34]. One study showed a delayed onset time of antiperistalsis, yet longer albeit non-significant duration of effect compared to IV administration of both agents [34]. This potential benefit has encouraged the implementation of a combination of IV and IM dosing into MRE protocols in order to further reduce image degradation due to motion artifacts, particularly in later sequences.

### 2.6. Challenges with MRE Preparation

In ideal scenarios, patients would be able to follow and complete all aspects of pre-MRE preparation and participate with the requirements of the scan. However, unique challenges such as claustrophobia and inability to consume the required volume of contrast can occur. In claustrophobic patients, coaching techniques to calm patients may be helpful. Anxiolytic medications can also be employed in the setting to aid the patient through the scan. In those unable to tolerate the required volume of oral contrast, optimization of imaging parameters and minimization of other artifacts is important. In our practice, careful interpretation of findings in bowel segments which are not well-distended is important to avoid over calling of findings. Reviewing acquired sequences while the patient is on the table to assess whether certain sequences require repeat imaging maybe considered on a case-by-case scenario.

## 3. Motion Artifact

Artifacts frequently degrade observed image quality, with population studies showing they occur in over 35% of small bowel segments assessed on MREs [35]. Broadly, two types of imaging artifacts manifest in cross-sectional bowel imaging: those occurring either directly as a consequence of the imaging protocol and technical acquisition (e.g., susceptibility artifacts from ferromagnetic material) or those occurring from patient-related factors (e.g., motion artifacts) during imaging acquisition [18]. Other examples of artifacts related to technical acquisition include from the presence of intraluminal air as a result of inadequate fluid distension. The presence of intraluminal air is known to influence the accuracy of DWI resulting in false-positive findings [25,36]. The focus of this review is on patient-related factors.

Motion artifacts may relate to a patient’s body movement, breathing during imaging acquisition, or bowel peristalsis movement, the latter of which is involuntary. International consensus statements aimed at standardising the MRE imaging protocol in patients with inflammatory bowel disease have been published in an effort to improve the performance and the ability for cross-comparison of MRE [19,37]. Despite this, the optimal MRE protocol to minimise the occurrence of artifacts in MRE assessment of CD remains unclear [11].

Motion artifacts are the cumulative effect of voluntary and involuntary patient movement during the image acquisition process. It typically manifests as blurring, ghosting, and/or smearing of the bowel wall and is thought to reduce sensitivity of disease activity detection in CD. Strategies to mitigate the effect of motion artifacts in general MRI include careful pre-scan explanation and instructions to patients of the need to remain still during image acquisition, ongoing coaching of regular breathing or breath holding for the appropriate sequences, providing comfortable on-table positioning, reducing image acquisition time, and occasionally providing sedation. However, the specific technical realities of MRE present several challenges that are yet to be fully addressed [38,39]. The unique barriers to minimising patient motion artifacts in MRE include the need for multiple breath holds over a scan period up to 45 min and peristaltic bowel wall movement which is currently mitigated, as described above, using IV and/or IM glucagon or hyoscine [34].

### 3.1. Current and Potential Strategies for Reducing Motion Artifact

#### 3.1.1. Impact of Contrast Delivery Mechanism on Motion Artifact (MRE vs. MR Enteroclysis)

Enteroclysis is a more invasive method of intraluminal contrast delivery via nasoenteric tube to the small bowel, and its utility has been limited by poor patient tolerance and the convenience of per oral consumption of contrast. Studies have compared the impact of oral versus enteroclysis delivery of contrast in small bowel MR on motion artifacts (and more broadly on image quality) [40,41] and are summarised in Table 1 and Table 2. Figurative representation of motion artifacts and bowel distension are provided in Figure 1, Figure 2 and Figure 3.

In a pilot study of 21 patients with known (or suspected) CD, Schreyer er al found no statistically significant difference in the presence of small bowel motion artifacts when comparing MRE with MR enteroclysis [41]. Similar findings were seen in a later study of 40 adult patients with known (or suspected) CD by Masselli et al. [40]. The latter, however, described a small but significant improvement in detection of superficial small bowel lesions with MR enteroclysis [40]; however, this was not explained by a difference in motion artifacts.

Both studies had limitations, in particular their small sample size. Furthermore, with no clear difference in motion artifacts across either modality, the magnitude of effect of motion artifacts on CD activity assessment could not be elucidated.

#### 3.1.2. Antispasmodics

The use of antispasmodics in MR enterography is recommended by the European Society of Gastrointestinal and Abdominal Radiology (ESGAR) and the American College of Radiology (ACR) guidelines to reduce peristalsis during image acquisition, thus potentially reducing wall blurring artifacts [19,57,58].

Motion artifacts may be further reduced by refinements in antiperistalsis administration such as split dosing and IM administration of antispasmodics [34,59,60]. IV Buscopan and glucagon were both shown to have to a have faster mean antiperistalsis onset time than the equivalent IM dose. There was no significant difference in duration of aperistalsis between IV and IM delivery with equivalent dosing [34]. Hence, a short time between administration of the IV aperistalsis agent and commencement of image acquisition is integral to acquiring bowel wall images that are less likely to be affected by motion artifact. A recent study showed that split dose IV Buscopan given prior to dynamic cine sequences significantly reduced the mean number of peristalsing small bowel loops and therefore may be a useful antiperistalsis delivery mechanism in reducing bowel motion artifact [59].

In a head-to-head trial, IV glucagon appeared more likely to achieve aperistalsis with a shorter time of onset compared to IV Buscopan [60]. Despite this, both agents appear to provide reliable antiperistalsis effect which may reduce the impact of motion artifact and improve the overall performance of MRE in CD detection activity.

#### 3.1.3. Breath Holding

Breath holding is required for multiple sequences obtained during MRE with free breathing limited mostly to DWI sequences. An inability to adequately hold a breath will inevitably cause motion artifacts related to movement of the diaphragm and subsequently abdominal organs, including the small and large bowel. Whilst widely performed, studies quantifying the true impact of breath holding on motion-artifact-related degradation in MRE are lacking.

Breath holding can be performed at end-expiration and inspiration with protocols developed according to local expertise. In a study relating to T1 Abdominal/Hepatic MRI, Vu et al. demonstrated a significant reduction in motion-related artifacts with end-expiration compared to end-inspiration in contrast-enhanced and unenhanced sequences [61]. Future studies assessing end-inspiration in MRE to determine its impact on motion artifacts and subsequent CD activity assessment are needed.

Current strategies to reduce motion artifacts relating to breath holding are limited to pre-MRE coaching by the MRI technologist performing the scan. These include the provision of clear instructions, practicing breath holding with patients as well as positive reinforcement during the scan. These instructions appear integral to patients remaining still for all types of MRI studies.

#### 3.1.4. Positioning—Prone vs. Supine

Traditionally, MRE is performed in a prone position as the anterior abdominal wall pressure appears to reduce to the volume of the peritoneal cavity [62]. The reduction in peritoneal cavity volumes appears to enhance separation of the small bowel and reduce abdominal movement which subsequently reduces the imaging sections needed to complete the MR sequences [62]. No study has directly compared the difference in observed motion artifacts in a prone versus a supine position. Cronin et al. conducted a study of 40 patients with known or suspected small bowel abnormalities who underwent a total of 62 MREs in both prone and supine position and found no significant difference in lesion detection or characterisation [63]. Further studies are needed to confirm the optimal position for image acquisition in patients with Crohn’s disease undergoing MRE, and both prone and supine positions are considered appropriate for MRE [19].

## 4. Definition and Grading of Motion Artifact in MRE Performed for Small Bowel and/or Crohn’s Disease Activity Assessment

Adult and paediatric MRE studies that have included motion artifact grading in the assessment of Crohn’s disease via MRE are summarised in Table 1. Due to a lack of clear definition, each study used different measurements of motion artifacts that were arbitrarily based upon local radiologist expertise.

Approaches to motion artifact grading include individual segmental scoring of motion artifact severity and combined quality grading with bowel distension (see Table 1). Individual grading has been performed with 3, 4, and 5-point scales with the level of severity of motion artifacts and the impact of diagnostic capacity as descriptors. Examples include Masseli et al. who utilised a 5-point scale which was segmentally graded in the following manner: (1) nondiagnostic images, (2) images with numerous artifacts, (3) diagnostic with few artifacts, (4) diagnostic with good quality, and (5) diagnostic images with excellent quality [40]. This categorisation, however, relies on a subjective approach by the reporting radiologist/s, may be challenging to reproduce, and may lead to unsatisfactory levels of interobserver variability. Studies have used clarity of the bowel wall as a marker of motion artifacts. Rieber et al. used a 3-point scale as per the descriptors: (1) Differentiation of the small bowel wall is impossible, (2) Differentiation of the small bowel wall is possible, but accurate assessment of the small bowel wall is impossible due to blurring, and (3) Clear differentiation of the small bowel wall with an accurate determination of the thickness of the small bowel wall is possible. This objective approach appears to provide a stepwise, logical method to grading motion artifacts though universal acceptance of this grading is lacking [46].

Finally, combined grading of bowel distension and motion has also been proposed. Koplay et al. used a 4-point scale incorporating both bowel distension and motion artifacts in the following manner: (1) excellent luminal distention without artifacts, (2) good luminal distention and mildly artifacted, (3) artifacted and inadequate but assessable in terms of CD, and (4) poor/nondiagnostic with CD assessment not possible [35]. This approach of combined grading of bowel distention and motion artifacts to formulate a quality score assumes, however, that both parameters contribute equally to quality. Moreover, it is yet to be shown whether suboptimal distension or the presence of a motion artifact result in decreased sensitivity in CD activity assessment and whether one parameter exerts a more significant impact on CD assessment than the other. The approaches to bowel distension grading will be discussed in subsequent sections.

To our knowledge, no consensus statements defining quality, adequacy, and specific parameters of motion artifacts in MRE have been developed. As a result, there is significant heterogeneity of the definition of motion artifacts in studies exploring specific interventions designed to reduce motion artifacts in MRE, and the comparison of different interventions is problematic.

## 5. Bowel Distention

Achieving adequate bowel distention is known to be critical to the successful execution of all small bowel cross-sectional imaging modalities. Indeed, disease activity interpretation in small bowel CD is particularly susceptible to inadequate distension given the resultant inability to detect subtle yet clinically significant lesions in this context.

Real-world practice of MRE performance varies across centres according to local fine-tuning of protocols. Previous studies have examined multiple aspects of oral contrast ingestion, such as type, volume, and timing relative to the image acquisition, and will be discussed.

### 5.1. Current and Potential Strategies for Improving Bowel Distention

#### 5.1.1. Impact of Contrast Delivery on Bowel Distension (MRE vs. MR Enteroclysis)

Few studies have sought to directly assess the impact of the delivery mechanism of oral contrast on bowel distension and subsequent CD activity assessment. Theoretically, nasoenteric tube insertion (MR enteroclysis) should improve bowel distension given the capacity to administer a larger amount of contrast in a shorter duration. However, in the study by Schreyer et al., there were no significant differences in bowel distention in the terminal ileum and proximal small bowel between oral and enteroclysis methods of delivery [41]. Furthermore, as previously mentioned, there was no significant difference in disease activity assessment in both groups [41].

Similar findings were demonstrated in a larger study of 40 adult patients with CD by Negaard et al. Whilst a significant difference in bowel distension was elicited in the jejunum and proximal ileum, this was not the case in the terminal ileum (*p* = 0.13) [64]. Once again, there was no significant difference in CD activity detection across both cohorts [64]. More recently though, Masselli et al. found in a cohort of patients with CD in which bowel distension was superior with enteroclysis in all small bowel segments [40]. Whilst there was a significant difference in the detection of superficial lesions, there was no difference with respect to other disease activity findings across both modalities [40].

Despite these data, MRE remains far more widely performed than enteroclysis with consensus guidelines not favouring either modality [2]. However, based on the available literature, there may be a select role for MR enteroclysis specifically in suspected proximal small bowel and/or superficial disease. Moreover, none of the studies was performed in the era of the recently developed MRE activity indices for CD; thus, it remains unclear how these indices would be affected by suboptimal small bowel distension.

#### 5.1.2. Type of Oral Contrast Used

The role of oral contrast is both to distend and opacify the small bowel lumen to enhance distinction between the lumen and the bowel wall [16,17]. Multiple oral contrast agents are described in the literature, though consensus regarding the most preferable specific agent has not been reached [19]. Compared to positive and negative contrast agents, biphasic contrast agents have specific advantageous properties. These include providing a low signal/dark appearing lumen on T1 contrast enhanced sequences, which is essential to assess the bowel mucosa [28], and high signal/bright appearing lumen on T2 sequences to facilitate assessment of bowel wall, which is dark on these sequences [17].

Examples of currently used biphasic contrast agents include barium sulfate, mannitol, sorbitol, polyethylene glycol (PEG), water, methylcellulose, and locust bean gum (LBG) [28,35,40,41,46,51,54,65,66]. Given its rapid absorption from the jejunum and beyond, water is seldom used. Due to individual agent properties and other confounding variables, studies comparing various contrast agents and their capacity to achieve adequate bowel distension have produced conflicting results. In a head-to-head comparison between sorbitol and psyllium in MRE, Saini et al. found no significant difference in bowel distension within the small bowel [67]. In a subsequent study by Schmidt et al., mannitol was found to be superior then psyllium for distension of the ileum, including the terminal ileum [55].

More recently, Bhatnagar et al. found jejunal distension to be significantly better with mannitol compared to PEG, though the ileum and terminal ileum were similar [54]. In a head-to-head trial of 75 patients with known or suspected CD, PEG, barium sulfate, and a newly developed sugar alcohol (combination of mannitol and sorbitol) were found to be equivalent for distention, maximal diameter, and opacification of the small bowel when utilised for MRE [68]. These outcomes suggest that multiple biphasic oral contrast agents appear to provide adequate bowel distension with optimisation of each contrast agent, subject to adjustment of volume and timing of ingestion, achieving satisfactory levels of bowel distension. Thus, the choice of particular agents is often primarily based upon local availability, experience, and patient tolerance.

#### 5.1.3. Volume of Oral Contrast Used

The optimal volume of oral contrast ingestion to achieve adequate bowel distension on MRE is not clearly defined in the literature, yet in standard practice, volumes of 1000–1500 mL are most commonly used. Interestingly, Kinner et al. suggested that adequate small bowel distension can be achieved with merely 450 mL of oral contrast [69]. More recently, a larger study of 105 patients with CD suggested that adequate small bowel distension could be achieved with <1000 mL of mannitol or PEG-based oral contrast, though the minimal cut-off volumes of oral contrast required to obtain an interpretable MRE were not established [54].

A key weakness of both studies was the subjective grading of small bowel distension rather than a more objective quantification. In addition, neither study evaluated the impact of a particular ingested volume on the accuracy of disease activity assessment. Hence, there is an unmet need to investigate the relationships between oral contrast volumes, objective grading of bowel distension, and the accuracy of CD activity assessment.

In summation, an individualised, flexible approach regarding the volume of oral contrast required may be preferable to achieve optimal bowel distension. This is especially relevant to patients following a total colectomy, resections of long segments of small bowel, and/or those with stoma formation. Many centres now follow a tailored approach after review of initial sequences while the patient is on table to assess whether a delay in scanning or the need for more oral contrast is required, rather than a ‘one size fits all’ approach.

#### 5.1.4. Timing of Oral Contrast

Timing of oral contrast in relation to image acquisition has been the subject of multiple studies [51,56,66]. Generally, it is recommended that oral contrast ingestion start 45–60 min prior to MRE image acquisition [56]. For example in one study, optimal small bowel distension was achieved in all 16 healthy volunteers who commenced oral contrast ingestion 45–60 min prior to MRE image acquisition, yet a significant loss of distension was seen in the duodenal and proximal jejunal segments when oral contrast ingestion was extended to 75 min prior to image acquisition [66].

More recently, in a larger cohort study of 100 patients with established CD, a shortened post-ingestion period of 45 min was superior to 60 min for distension in the stomach, duodenum, jejunum, and total small bowel when administering 1600 mL of 2% mannitol. However, there was no significant difference in proximal and terminal ileum distension across both the 45 and 60 min cohort, which is particularly relevant given the predominant distribution of small bowel CD within the terminal ileum [51].

The individual properties of oral contrast agents may also affect the optimal timing of contrast ingestion. For instance, despite similar volumes and timing of oral ingestion, mannitol has been shown to achieve better jejunal distension than polyethylene glycol [54]. Furthermore, in patients with CD who have undergone significant lengths of small bowel resection(s) and/or stoma formation, there are minimal evidence-based protocols available. Expert consensus groups have suggested a shortened ingestion period of 30 min and/or stoma plugging to optimise bowel distension, but this is not widely incorporated into real-world practice [19]. Furthermore, patients with stoma formation are ideally imaged in a supine position to avoid the potential overfilling and leakage of stoma bags.

#### 5.1.5. Rectal Filling

Rectal filling with water has been performed in studies assessing the ability of MRE to also incorporate colonic CD assessment [50,56,65]. Along with improvement in large bowel distension, studies have demonstrated a significant improvement in small bowel distension with the application of a rectal enema/instillation of fluids via catheter and subsequent improved detection of CD-related changes in the small bowel [70]. Whilst clearly mitigating the potential procedural risks compared to ileocolonoscopy, the use of rectal water instillation when performing MRE or MR enterocolonography for CD assessment is currently not performed widely [19]. Factors limiting the usage include the extra overall scanning time as well as the reduced patient tolerability of rectal instillation of water [71].

#### 5.1.6. Prokinetic Agents

Prokinetic agents such as erythromycin and metoclopramide have been applied in MRE protocols [66] in an attempt to rapidly increase small intestinal volume by accelerating gastric emptying of oral contrast. In a study of healthy volunteers undergoing MRE, the application of 200 mg IV erythromycin did not enhance small intestinal contrast volumes compared with placebo [72]. However, a small but significant increase in ileal loop distension was observed, though given the study was performed in a healthy cohort, the impact on CD activity assessment is uncertain [72]. Overall, the authors concluded that the impact of prokinetics was minimal, which aligns with current practice as prokinetics are rarely used prior to MRE image acquisition.

## 6. Definition and Grading of Bowel Distention in MRE Performed for Small Bowel and/or Crohn’s Disease Activity Assessment

Adult and paediatric MRE studies that describe bowel distension grading in MR enterography/enteroclysis are summarised in Table 2. Due to a lack of clear, accepted definitions, each study used different measurements of bowel distension which were typically a combination of subjective and objective assessments developed per local radiologist expertise.

Commonly observed approaches to bowel distension grading include segmental qualitative scoring with 3-, 4- and 5-point scales, with some studies incorporating the impact of diagnostic capacity in the grading (Table 2). Examples include Absah et al. who utilised a 5-point scale from the worst, (1) luminal collapse compromising diagnostic interpretation through to the best, (5) excellent optimal bowel distention [42]. As for motion artifacts, this approach relies on subjective assessment by the reporting radiologist/s, thus potentially diminishing reproducibility interobserver reliability.

Another qualitative approach employed by several studies is to grade distension across an entire bowel segment rather than a single anatomical point. For instance, Bekendam et al. proposed the following grading system: 0 = no distension or collapsed segment (<25% of segment adequately distended), 1 = insufficient distension (25–50% of segment adequately distended), 2 = sub-optimal distension (50–75% of segment adequately distended), 3 = optimal distension (>75% of segment adequately distended). This method appears more robust than a single point measurement, particularly when assessing long segment, small intestinal CD [51]. Alternatively, bowel distension grading has been performed via combined assessment of artifacts and bowel distension as previously discussed in this review. However, quantitative description and grading of bowel distension in MRE and MR enteroclysis are lacking in the current literature.

To our knowledge, no consensus statements defining quality, adequacy, and specific parameters of bowel distension in MRE have been developed. As a result, there is significant heterogeneity in the definition of bowel distension as reflected in studies examining methods to improve small bowel distension in MRE.

## 7. Discussion

MRE is a widely used tool for the assessment of small CD. With CD treatment targets increasingly focused on normalising previously abnormal assessment findings, high quality, reproducible images are critical to ensuring accurate CD activity assessment and hence aiding clinical decision making. Similar to the application of quality metrics in multiple aspects of healthcare, these have also been considered in the evolution of MRE utilisation for the assessment of small bowel CD. Studies have hitherto focused on strategies to minimise the detrimental effects of motion artifacts and suboptimal bowel distension, the two-quality metrics most amenable to change in cross-sectional small bowel imaging. This review highlights the significant heterogeneity in MRE protocols, the descriptors of motion artifacts and bowel distension, and the approaches to mitigating these.

The main challenge to improving quality in MRE elucidated by this review is the lack of published data and/or expert consensus guidelines regarding definitions and grading of optimal versus suboptimal levels of each quality metric. As a consequence, widely disparate gradings have been applied to each metric across various studies as described in this review. Typically, therefore, the assessment of adequacy of bowel distension and motion artifacts is, at best, left to local radiologist expertise or at worst, ignored completely. Moreover, most studies utilise subjective rather than objective, reproducible analyses of image sequences which preclude the opportunity for cross-comparison of intervention strategies across different studies. Hence, it is problematic to determine the optimal type, volume, and timing of oral contrast used in MRE as well as other intervention strategies, including the use of anti-peristaltics and breath holding in addressing motion artifacts and/or bowel distension.

Accurate CD assessment is a prerequisite to any modality of disease activity assessment given that inaccuracies have significant downstream implications for therapeutic decision making. This is no different with MRE, where findings relevant to CD have significant clinical implications, such as affecting decisions regarding treatment escalation and choice or referral for surgical resection. However, there are scant real-world data defining the proportion of MRE scans performed that achieve adequate image quality to accurately determine CD activity. This has clear implications on MRE-based CD activity indices which assume consistent, widespread MRE image adequacy. Hence, MRE scans with quality limitations should be excluded from studies evaluating MRE performance and should at least be clearly stated to facilitate real world comparisons.

In terms of future directions, a logical stepwise approach would first involve the development of expert consensus-derived definitions of bowel distension and motion artifact grading with an objective, reproducible framework. It is important that these definitions are straightforward to apply and thus subsequently have the potential to become a routine adjunct to best practice MRE reporting. Aside from the academic benefits to comparing interventions in MRE, this would also empower clinicians with a more nuanced interpretation of an individual MRE’s suitability on which to confidently base clinical decisions.

## 8. Take Home Messages

Adequate bowel distension and minimal motion artifacts are a critical prerequisite for high-quality MRE images;Optimal bowel distension can be achieved with 1000 mL of oral contrast given 45 to 60 min prior to image acquisition;Reduction of motion artifacts can be achieved through coaching patients with breath-holding strategies and through the timed use of antispasmodics;Antispasmodic properties can be manipulated to achieve minimal motion artifacts by utilising both IV and IM routes of delivery;Description of grading of both bowel distension and motion artifacts are poorly described in the literature but are a critical part of MRE reporting in CD;Consensus definitions of both bowel distension and motion artifacts in MRE are needed.

## Figures and Tables

**Figure 1 diagnostics-12-02533-f001:**
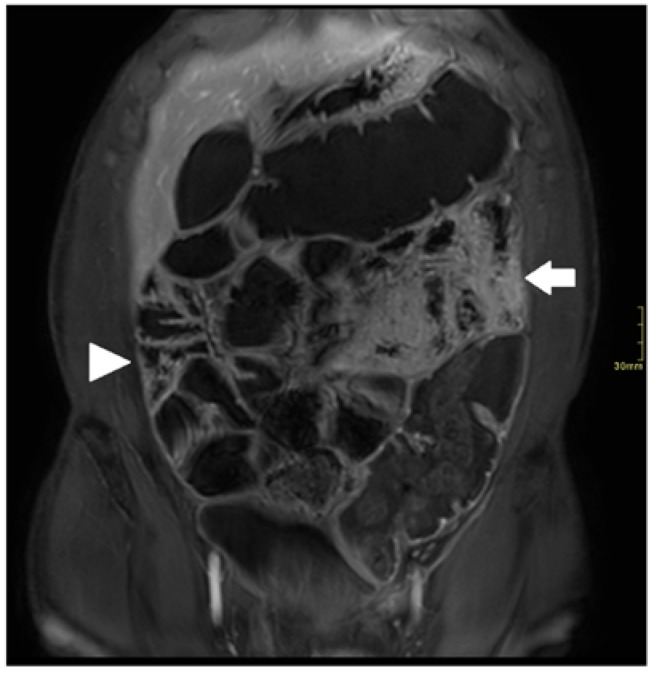
Coronal fat-suppressed T1 post-contrast sequence of a 61-year-old male with small bowel CD; suboptimal distension and marked motion artifacts in proximal small bowel (arrow); optimal distension and minor motion artifacts in distal small ball (arrowhead).

**Figure 2 diagnostics-12-02533-f002:**
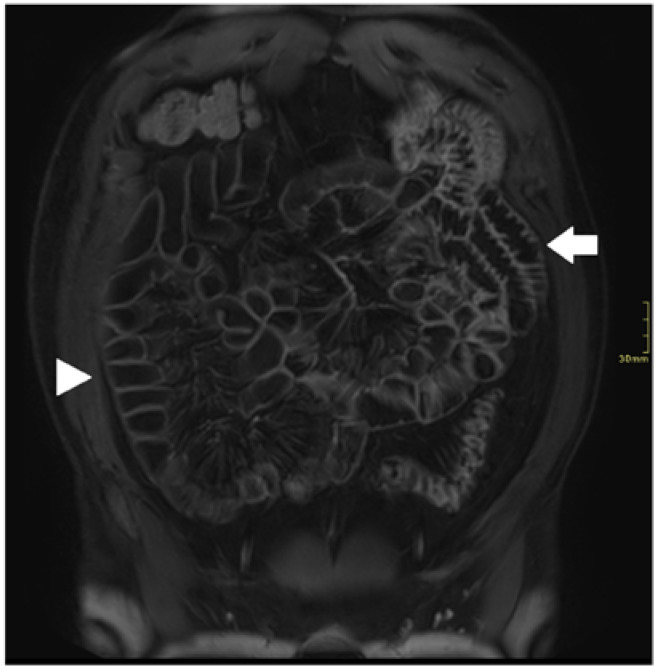
Coronal fat-suppressed T1 post-contrast sequence of a 35-year-old male with small bowel CD; optimal distension and no motion artifacts in proximal (arrowhead) and distal (arrow) small bowel.

**Figure 3 diagnostics-12-02533-f003:**
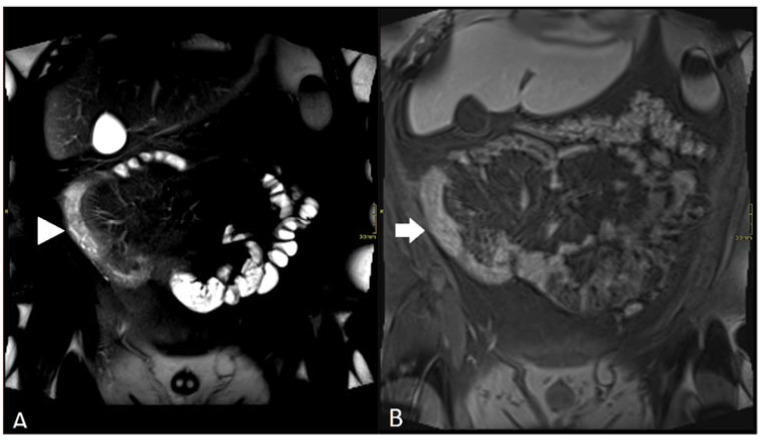
Sequence of a 23-year-old male with small bowel CD: (**A**) coronal fat-suppressed T2 sequence showing long segment of inflammatory disease in distal ileum (arrowhead); high T2 signal in bowel wall and adjacent fat; (**B**) coronal fat-suppressed T1 post-contrast sequence showing poor distension and marked motion artefact in distal ileum (arrow) leading to underestimation of degree of inflammation.

**Table 1 diagnostics-12-02533-t001:** Summary of MR enterography studies describing motion artifacts.

Study	CD vs. Non-IBD	Adult vs. Paediatric Cohort	*n*	Aim(s) of Study	Type/Volume Ingestion (mL)/Time	Antiperistalsis Agent/Dose	Bowel Segment	MRI Strength	Description of Motion Artifact	Study Outcome Relevant to Motion Artifact
Absah et al. [42]	CD	P	70	Evaluate image quality, oral contrast administration and bowel distention, side effects, and performance estimates of MRE	Barium sulfate/600 mL/2 h Barium sulfate/1350 mL/2 h if weight > 27.2 kg If volume not tolerated, water and/or juice offered as well.	IV glucagon/0.006 mg/kg	J I AC TC DC RS	1.5 T	Respiratory and peristalsis-related artifact grading: 1 = uninterpretable 2 = moderate to severe artifacts resulting in markedly decreased diagnostic confidence 3 = moderate artifacts resulting in moderately decreased diagnostic confidence 4 = minimal artifacts not affecting diagnostic confidence 5 = no artifacts with excellent image quality). >4 was acceptable	Mean image artifact quality score for unenhanced pulse sequences was 4.7 ± 0.5 which was significantly greater than gadolinium-enhanced sequences at 4.1 ± 0.8 (*p* < 0.0001; mean difference, 0.66; 95% CI, 0.51–0.81). Respiratory motion artifacts were highest in coronal 2D fast spoiled gradient-recalled echo sequences.
Bosemani et al. [43]	CD	P	34	Evaluate developed MRE protocol	Barium sulfate/10 mL per kg/60 min	IV glucagon/0.5 mg if <20 kg/1 mg if >20 kg	D J I TI C R AnalC	1.5, 3 T	Overall bowel grading of study: 0 = nondiagnostic, extensive motion artifacts 1 = diagnostic, moderate motion artifacts 2 = highly diagnostic, no motion artifacts	Overall grading of study: 0 = 1/34 (2.9%) 1 = 7/34 (20.6%) 2 = 26/34 (76.5) 76.5% of assessed image sequences had no motion artifacts
Borthne et al. [44]	Suspected CD	P	43	Determine the diagnostic accuracy of small bowel MRI using mannitol as the contrast agent	Mannitol/weight based/60 min Median 300 mL (range 200–500 mL)	IV Buscopan/20 mg	D J I TI AC TC LC	1.5 T	Grading: 0 = blurred bowel wall 1 = only portion of the bowel wall segment visible 2 = wall of the segment seen but not in its entire length 3 = wall clearly seen in its entire length 4 = excellent delineation of the wall	Median Score D = 3 J = 3 I = 4 TI = 4 AC = 4 TC = 2 LC = 0 Median image quality was evaluated as very good or excellent from the jejunum to the ascending colon
Lawrance et al. [45]	Mixed	A	114	Evaluation of MRE vs. MR enteroclysis	MRE: Polyethylene glycol/1000 mL/20 min MR enteroclysis: polyethylene glycol/1000–2000 mL/immediately prior to scan	IV Buscopan/10 mg	Prox. SB Distal SB	1.5 T	Present if images diagnostically impaired on both sequence planes through a region	Outcome: Image artifacts were present more frequently in MR enteroclysis than in MRE (29.2% vs. 18.4%) but non-significant (*p* = 0.30)
Rieber et al. [46]	Inflammatory or tumorous bowel disease	A	50	Diagnostic capacity of MR enteroclysis with positive vs. negative contrast media	Barium suspension and methyl hydroxycellulose/1400 mL/immediately prior to scan Plus, positive, or negative contrast	IV Buscopan/20 mg	J PTI TI Ce	1.5 T	Grading: 1 = differentiation of small bowel wall is impossible 2 = differentiation of small bowel wall is possible but accurate assessment of small bowel wall is impossible due to blurring 3 = clear differentiation of the small bowel wall with an accurate determination of the thickness of the small bowel wall.	Not assessed
Koplay et al. [35]	Known or presumed CD	A and P	153	Diagnostic accuracy and image quality of MRE using oral mannitol solution vs. colonoscopy	3% mannitol/1300–1500 mL/60 min	IV Buscopan/20 mg or weight based if <50 kg	D J I TI	1.5 T	Combined grading with bowel distension: 1 = excellent luminal distention, without artifacts 2 = good luminal distention, mildly artifacted 3 = artifacted, inadequate but assessable in terms of CD 4 = poor/nondiagnostic, cannot be assessed.	Mean quality score per segment D = 1.92 J = 1.6 I = 1.12 TI = 1.15 37% of assessed segments were affected by motion artifacts. Lowest quality seen in duodenum and jejunum.
Grand et al. [47]	Mixed	A	26	Diagnostic capacity of CTE and MRE without antiperistalsis agents	Barium sulfate/900 mL/60 min	Nil	TI C	1.5 T	Grading: 0–10 (10 highest) of exam quality (combining presence/absence of motion artifacts and adequacy of bowel distention)	Reader 1: MRE quality 9.0/10 (6–10) Reader 2: MRE quality 7.2/10 (6–10) Quality of MRE was lower compared to CTE but no difference in diagnostic accuracy
Dagia et al. [48]	CD	P	42	Diagnostic accuracy of 3 T MRE	Sorbitol/1000–1500 mL/60 min 36/42 MR enterography 6/42 MR enteroclysis	IV Buscopan/weight based up to 20 mg	J + PTI TI Ce + C	3 T	Image quality affected by respiratory and motion artifacts vs. not affected, evaluated for each sequence	T2 sequences: 17 to 21% affected T1 post-contrast sequences: 42 to 59% affected
Frøkjær et al. [28]	Likely CD	A	36	Diagnostic accuracy of MRE compared to conventional enteroclysis	Plum juice plus psyllium husk/1000 mL/150 min	IV Buscopan/20 mg	TI Ce	1.5 T	Grading by addition of score for artifacts and distension (max score of 6): 0 = no distension 3 = maximal distention 0 = severe artifacts 3 = no artifacts	Mean quality score: 4.49 ± 0.8 No sub-analysis of motion artifacts performed
Siddiki et al. [49]	CD	A	33	Diagnostic accuracy of MRE vs. CTE	Barium sulfate + water/1850 mL/60 min	IV glucagon/0.5 mg	D Proximal J Distal J PTI I TI	1.5 T	Grading: 1 = uninterpretable 2 = moderately severe artifacts, markedly diminished confidence 3 = moderate artifacts, mild to moderate decrease in reader confidence 4 = minimum artifacts, no effect on confidence 5 = excellent quality	Mean MRE quality score of 4.7 (Mean CTE quality score of 5) MRE showed significantly lower quality vs. CTE (*p* = 0.005)
Schreyer et al. [50]	Known or presumed CD	A	22	Diagnostic accuracy of MR colonography vs. conventional colonoscopy	Nil Bowel preparation completed day prior/1.5 L of gadolinium and water mixture inserted per rectally	IV Buscopan/40 mg	TI Ce AC TC DC SC R	1.5 T	Combined grading with bowel distension: 0: >50% of the segment was not adequately distended or image impression was disturbed by motion artifacts 1: segment distension from 50% to 80% without major motion artifacts air quality 2: segment distended more than 80% without any image artifacts	Colonic segments and TI assessed. Did not assess remainder of small bowel. Grading outcome: 0 = 2.6% of segments 1 = 20.1% of segments 2 = 77.3% of segments scanned MR colonography provided good quality scans from the TC to the rectum.
Schreyer et al. [41]	Known or presumed CD	A	21	Diagnostic accuracy of MR enteroclysis vs. MRE	MR enteroclysis: barium sulfate with methylcellulose/1990 mL/immediately pre-scan MRE: pineapple juice with methylcellulose/2000 mL/120 min	IV Buscopan/40 mg	Proximal SB TI	1.5 T	Grading: 1 = nondiagnostic images 2 = images with numerous artifacts 3 = diagnostic images with few artifacts 4 = diagnostic images with good quality 5 = diagnostic images with excellent quality	MRE: Mean quality proximal SB: 4.68 ± 0.58, TI: 4.25 ± 0.91 MR enteroclysis Mean quality proximal SB: 4.35 ± 0.88, TI: 4.75 ± 0.64 No significant differences in motion artifacts between MRE and MR enteroclysis
Masselli et al. [40]	CD	A	40	Diagnostic accuracy of MR enteroclysis vs. MRE and CT enteroclysis	MR enteroclysis: polyethylene glycol/1600–2000 mL/immediately prior to scan MRE: polyethylene Glycol/1800 mL/50 min	IV Buscopan/20 mg	J PTI TI	1.5 T	Grading: 1 = nondiagnostic images 2 = images with numerous artifacts 3 = diagnostic images with few artifacts 4 = diagnostic images with good quality 5 = diagnostic images with excellent quality	MRE: Mean quality score of 4.4 MR enteroclysis: Mean quality score of 4.5 No statistically significant differences between MRE and MR enteroclysis (*p* = 0.13)
Bekendam et al. [51]	CD	A	100	Comparison of bowel distension and image quality based upon time prior to scan in which oral contrast ingested	2% mannitol/1600 mL/45 min vs. 60 min	IV Buscopan/20 mg	S D J I TI SB	1.5 T	Grading: 0 = nondiagnostic quality 1 = diagnostic quality with numerous artifacts 2 = diagnostic with a few artifacts 3 = diagnostic with no artifacts	All scans considered diagnostic quality Mean quality scores for oral contrast over 45 min for two readers: 2.8 and 2.2 Mean quality scores for oral contrast over 60 min for two readers: 2.8 and 2.1

Abbreviations: CD: Crohn’s Disease; P: paediatric; A: adult; IBD: inflammatory bowel disease; J: jejunum; PTI: proximal terminal ileum; TI: terminal ileum; I: ileum; AC: ascending colon; TC: transverse colon; DC: descending colon; RS: rectosigmoid; SC: sigmoid colon; C: colon; R: rectum; AnalC: anal canal; LC: left colon; SB: small bowel; Ce: Caecum; CTE: computed tomography enterography; MRE: magnetic resonance enterography; MR: magnetic resonance; IV: intravenous.

**Table 2 diagnostics-12-02533-t002:** Summary of MR enterography studies describing bowel distension.

Study	CD vs. Non IBD	Adult vs. Paediatric Cohort	*n*	Aim(s) of Study	Type/Volume Ingestion (mL)/Time (min)	Antiperistalsis Agent/Dose	Bowel Segment	MRI Strength	Description of Bowel Distension	Study Outcome Relevant to Bowel Distension
Absah et al. [42]	CD	P	70	Evaluate image quality, oral contrast administration and bowel distention, side effects, and performance estimates of MRE	600 mL/2 h 1350 mL/2 h if weight > 27.2 kg If volume not tolerated, water and/or juice offered as well.	IV glucagon/0.006 mg/kg	J I AC TC DC RS	1.5 T	Grading: 1 = luminal collapse compromising diagnostic interpretation 2 = markedly suboptimal bowel distention resulting in markedly decreased diagnostic confidence 3 = moderately suboptimal distention resulting in moderately decreased diagnostic confidence 4 = good but suboptimal bowel distention not affecting diagnostic confidence 5 = excellent optimal bowel distention	Poor bowel distension resulting in reduction in image quality. Age was an independent predictor of volume of ingested contrast(*p* = 0.009)
Bosemani et al. [43]	CD	P	34	Evaluation of developed MRE protocol	Barium sulfate/60 min/10 mL per kg	IV glucagon/0.5 mg if <20 kg IV glucagon/1 mg if >20 kg	D J I TI C R AnalC	1.5 and 3 T	Grading 0 = poor distension 1 = mild distension 2 = moderate distension 3 = excellent distension	Grading (3/excellent): D = 29/34 (85.3%) J = 29/34 (85.3%) I = 31/34 (91.2%) TI = 21/34 (61.8%) Caecum = 12/34 (44.4%) Colon = 17/34 (77.3%) Rectum = 7/34 (36.8%)
Borthne et al. [44]	Suspected	P	43	Determine the diagnostic accuracy of MRI of the small bowel using mannitol as the contrast agent	Mannitol/weight based/60 min Median 300 mL (range 200 to 500 mL)	IV Buscopan/20 mg	D J I TI AC TC LC	1.5 T	Grading: 0 = poor 1 = moderate 2 = good 3 = very good 4 = excellent	Distension outcome (>grade 3): D 47.1% J 58.5% I 79.2% TI 77.3%
Lawrance et al. [45]	Mixed	A	114	Evaluation of MRE vs. MR enteroclysis	MRE: polyethylene glycol/1000 mL/20 min MR enteroclysis: polyethylene glycol/immediately prior to scan/1000–2000 mL	IV Buscopan/10 mg	Proximal SB Distal SB	1.5 T	Good distension was defined as: luminal fluid present within the bowel lumen allowing clear visualisation of both endoluminal surfaces Any other was defined as poor	Grading outcome (good): Proximal SB: MRE 65.3%, MR enteroclysis 84.6% Distal SB: MRE 97.8% MR enteroclysis 95.4% Proximal SB distension was overall more likely to be suboptimal in MRE than in MR enteroclysis (OR = 4.365, 95% CI = 1.62–11.77, *p* = 0.0036)
Frøkjær et al. [28]	Likely CD	A	36	Diagnostic accuracy of MRE compared to conventional enteroclysis	Plum juice plus psyllium husk/1000 mL/150 min	IV Buscopan/20 mg	TI Ce	1.5 T	Grading by addition of score for artifacts and distension (max score of 6): 0 = no distension 3 = maximal distention 0 = severe artifacts 3 = no artifacts	Mean quality score: 4.49 ± 0.8No sub-analysis of bowel distension performed
Schreyer et al. [50]	Known or presumed CD	A	22	Diagnostic accuracy of MR colonography compared to conventional colonoscopy	Nil Bowel preparation completed the day prior/1.5 L of gadolinium and water mixture was inserted per rectally	IV Buscopan/40 mg	TI Ce AC TC DC SC R	1.5 T	Combined grading with bowel distension: 0: >50% of the segment was not adequately distended or image impression was disturbed by motion artifacts 1: segment distension from 50% to 80% without major motion artifacts air quality 2: segment distended more than 80% without any image artifacts	Colonic segments and TI assessed. Did not assess remainder of small bowel. Grading outcome: 0 = 2.6% of segments scanned 1 = 20.1% of segments scanned 2 = 77.3% of segments scanned MR colonography provided good quality scans from the TC to the Rectum.
Ajaj et al. [52]	Healthy volunteers	A	10	Comparison of 4 different volumes of oral contrast agent with 2.5% mannitol and 0.2% LBG volume	2.5% mannitol and 0.2% LBG/1500 mL, 1200 mL, 1000 mL, 800 mL/45 min prior to scan with IV erythromycin	Nil	Eight small bowel loops spaced between the jejunum and ileum	1.5 T	Grading: 0 = very poor 1 = poor 2 = fair 3 = good 4 = excellent	Mean grade of distension: 1500 mL: 3.4 1200 mL: 3.2 1000 mL: 3.1 800 mL: 2.0 800 mL was statistically inferior to higher volumes (*p* > 0.05) 1200 and 1500 mL had more side effects.
Ajaj et al. [53]	Healthy volunteers	A	12	Comparison of 4 different oral contrast agents combined with 0.2% LBG	Mannitol 2.5%, sorbitol 2.5%, sorbitol 2.0%, sorbitol 1.5%/1500 mL/40 min pre-scan with IV erythromycin	Nil	Eight small bowel loops spaced between the jejunum and ileum	1.5 T	Grading: 0 = very poor 1 = poor 2 = fair 3 = good 4 = excellent	Quantitative small bowel distension was statistically significant for mannitol compared with sorbitol 2.5% and sorbitol 1.5%. The sorbitol 2.0% solution showed no statistically significant difference to mannitol. No statistically significant difference in loop diameter was found between ileum and jejunum for all four contrast solutions.
Bekendam et al. [51]	CD	A	100	Comparison of bowel distension and image quality based upon time prior to scan in which oral contrast ingested	2% mannitol/1600 mL/45 vs. 60 min	IV Buscopan/20 mg	S D J I TI SB	1.5 T	Grading: 0 = no distension or collapsed segment (<25% of segment adequately distended). 1 = insufficient distension (25–50% of segment adequately distended) 2 = sub-optimal distension (50–75% of segment adequately distended) 3 = optimal distension (>75% of segment adequately distended)	Statistically significant (*p* < 0.05) improvement in distension in the 45-min cohort within the stomach, jejunum, and total SB. No statistical differences for ileum and terminal ileum between two protocols.
Koplay et al. [35]	Known or presumed CD	A and P	153	Diagnostic accuracy and image quality of MRE using oral mannitol solution vs. colonoscopy	3% mannitol/1300–1500 mL/60 min	IV Buscopan/20 mg or weight based if <50 kg	D J I TI	1.5 T	Combined grading with bowel distension: 1 = excellent luminal distention, without artifacts 2 = good luminal distention, mildly artifacted 3 = artifacted, inadequate but assessable in terms of CD 4 = poor/nondiagnostic, cannot be assessed.	Mean quality score per segment D = 1.92 J = 1.6 I = 1.12 TI = 1.15 Best dilatation seen in ileal and terminal ileum
Schreyer et al. [41]	Known or presumed CD	A	21	Diagnostic accuracy of MR enterocylsis compared to MRE	MR enteroclysis: barium sulfate with methylcellulose/1990 mL/immediately prior to scan MRE: pineapple juice with methylcellulose/2000 mL/120 min	IV Buscopan/40 mg	Proximal SBTI	1.5 T	Objective grading: 1 = 20% of the small bowel adequately distended 2 = 20–40% adequately distended 3 = 40–60% adequately distended 4 = 60–80% adequately distended 5 = excellent distention, 80% distended	MRE: Mean quality proximal SB: 4.68 ± 0.58, TI: 4.25 ± 0.91 MR enteroclysis Mean quality proximal SB: 4.35 ± 0.88, TI: 4.75 ± 0.64 No difference in distension in proximal SB and TI between MR enteroclysis and MRE. No significant difference in CD activity assessment across MR enteroclysis and MRE.
Bhatnagar et al. [54]	CD	A	105 (68 mannitol cohort, 37 in PEG cohort)	PEG vs. mannitol for improved bowel distension	PEG, mannitol/1500–2000 mL/45–60 min prior to scan	Unknown	D J I TI RC	unknown	Grading: 0 = very poor distension 1 = poor distension 2 = fair distension 3 = good distension 4 = excellent distension	Per patient distension quality was similar between agents. Jejunal distension was significantly better with mannitol compared to PEG. No difference if <1 L or >1 L on ileum/terminal ileum distension but jejunum better if >1 L.
Schmidt et al. [55]	Non-IBD	A	45	Comparison of 3 contrast agents to determine which provides the best small bowel distension in MRE/MR enteroclysis	MR enteroclysis: pirenzepine day prior. tylose/1500 mL/immediately prior to scan MRE: LBG in mannitol/1500 mL/90 min MRE: psyllium in water/1500 mL/270 min	IV Buscopan/10 mg	D TI Ce	1.5 T	Grading: 1 = very good 2 = good 3 = moderate 4 = poor 5 = impossible Diameter of small bowel also assessed	Tylose via MR enteroclysis was superior to psyllium in water and LBG in mannitol for image quality in the duodenum and proximal jejunum (*p* = 0.030) LBG in mannitol was superior to the other methods for distension of the ileum and terminal ileum(*p* = 0.031).
Tkalčić et al. [56]	Mixed	A	164 CD 53 healthy subjects	Optimal time to start oral contrast ingestion prior to MR enterocolonography. Whether rectal water enema instillation and use of spasmolytics improves bowel distension	0.25% mannitol/1000–1500 mL/60 min Starting 90, 75, 60, 45 and 30 min pre-MR enterocolonography. Also had bowel preparation with PEG day prior	IV Buscopan/20–40 mg	J I TI C LB	1.5 T	Grading: 1 = several collapsed bowel segments or longer segment without luminal content 2 = isolated collapse of short bowel segment 3 = fair bowel distension with luminal content visible in all parts of dependent segment 4 = good bowel distension with a potential for maximal distension 5 = excellent bowel distension	Outcomes: 30–75 min ingestion time provided similar bowel distension with a significant(*p* <0.05) reduction in the 90 min cohort. Application of spasmolytics: all segments of the small bowel showed significantly better distension (*p* < 0.05). Application of water enema had significant influence on all segments of large and small bowel distension, (*p* < 0.05) except for the jejunum in the CD group.

Abbreviations: LBG: locust bean gum; S: stomach; PEG: polyethylene glycol; LB: large bowel; CD: Crohn’s Disease; P: paediatric; A: adult; IBD: inflammatory bowel disease; J: jejunum; PTI: proximal terminal ileum; TI: terminal ileum; I: ileum; AC: ascending colon; TC: transverse colon; DC: descending colon; RS: rectosigmoid; SC: sigmoid colon; C: colon; R: rectum; AnalC: anal canal; LC: left colon; SB: small bowel; Ce: caecum; CTE: computed tomography enterography; MRE: magnetic resonance enterography; MR: magnetic resonance; PET: positron emission tomography; IV: intravenous; L: litre.

## Data Availability

Not applicable.

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
