# Peer review of "Challenges and Strategies to Optimising the Quality of Small Bowel Magnetic Resonance Imaging in Crohn’s Disease"

_diagnostics, 2022, doi:10.3390/diagnostics12102533_

Round 1
Reviewer 1 Report
This article represents a great deal of work on difficult issues for which there is no international agreement.
I completely agree that there is a lack of standardization of the MRE exam and its course.
Thus, the objectives are relevant and interesting to your readers.
Introduction:
- Is there a place for the videocapsule in the CD evaluation?
- An important parameter to discuss is also the presence and importance of air in the gut, a source of artifacts when using DWI.
2.4 IV contrast: the impact and interest of gadolinium injection has been questioned in many articles.
- Diffusion-weighted magnetic resonance without bowel preparation for the detection of colonic inflammation in inflammatory bowel disease.Oussalah A, Laurent V, Bruot O, Bressenot A, Bigard MA, Régent D, Peyrin-Biroulet L.Gut. 2010 Aug;59(8):1056-65. doi: 10.1136/gut.2009.197665. Epub 2010 Jun 4.PMID: 20525970
- Prospective evaluation of free-breathing diffusion-weighted imaging for the detection of inflammatory bowel disease by magnetic resonance enterography in the pediatric population. Céline Dubron 1, Freddy Avni 1, Nathalie Boutry 1, Dominique Turck 2, Alain Duhamel 3, Elisa Amzallag-Bellenger 1 PMID: 26838954 PMCID: PMC4846210 DOI: 10.1259/bjr.20150840
2.5 antiperistaltic agent: Dilman and colleagues reported the benefit of glucagon for intestinal distension, but the clinical impact was unknown.
Dillman JR, Smith EA, Khalatbari S, Strouse PJ (2013) I.V. glucagonuse in pediatric MR enterography: effect on image quality, examination time, and patient tolerance. AJR Am J Roentgenol 201:185-189
2.6 motion artifacts: air has to be discussed as already mentioned
3.1.2 Antispasmodics: quite similar chapter redundant to 2.5
3.1.4: positioning decreases the volume of the peritoneal cavity by mainly trying to divide the bowel loops by pressing on the belly and reducing belly movement.
5 bowel lumen distension
Already mentioned in the previous paragraph and in Chapter 2 make the presentation confusing. The discussion on the classification of combinations (distension/movement) could be addressed in another paragraph.
5.1.1 Impact on contrast delivery already discussed in 3.1.1.
- The value of the ADC measurement is not discussed
- The diagnostic power of the high parietal signal with the use of a high b-value related to bowel distension is not discussed.
5.1.3 volume: "An individualized approach after review of the initial sequences while the patient is on the table to assess whether a delay in scanning or the need for more oral contrast is necessary is therefore recommended": this is already done in many, if not all, centers dealing with MRE.
7: Other quality parameters that may be altered in small bowel imaging are the MR sequences used and the magnet power.
The bibliography and table include colonography (Schreyer) which is a different focus.
Table 2: The title is incorrect - this is not motion artifacts but bowel distension.
Reviewer 2 Report
The authors extensively described challenges and strategies to optimizing the quality of MRE in patients with CD.
The review is well-written, exploring most of the critical issues of MRE.
Nevertheless, some points need to be addressed before possible acceptance.
MAJOR POINTS
- One could expect that a review on MRE includes some MR images illustrating the concepts described in the paper, e.g., cases of various bowel distention and types of artifacts.
- Paragraph 2.1 at pag.2 includes a generic description of the MRE protocol. Consider adding the rationale of preferring certain sequence types to lessen artifacts. A synthetic table may help to achieve this commitment.
- The authors should select some take-home messages/bullet points, resuming the content of each paragraph. Consider adding a dedicated Table or Figure.
MINOR POINTS
- Paragraph 2.2 lacks references. I am not sure to understand the effect of fasting on bladder over-filling. The potential impact of fasting on luminal signal homogeneity on MRE should be discussed.
- Paragraph 2.5 lacks references.
- The discussion should be shortened (please refer to major point #3).
- The title of Table 2 is incorrect.
- Consider moving both Tables 1 and 2 into supplementary material while resuming data on distension and artifacts reduction through the pro&cons of different approaches.
- Consider adding a short paragraph suggesting strategies to be applied in “difficult” patients, e.g., claustrophobic or intolerant to bowel distention, to minimize artifacts while optimizing the MRE protocol not to miss relevant findings (are there any sequences to be preferred over others?).
- The lack of MRE images that, when added, would provide significant added value to the manuscript;
- The lack of MRE protocol details, such as sequences' choice and adaptions, can significantly impact artifacts and, in some cases, counterbalance a suboptimal bowel distention.
Round 2
Reviewer 2 Report
The authors properly addressed all the issues I have raised.